# Enarodustat suppresses thymic stromal lymphopoietin expression via hypoxia-inducible factor-mediated c-Jun N-terminal kinases dephosphorylation

Ryosuke Segawa[1,2*], Makiko Yagisawa[1], Chihiro Miyata[1], Noriyasu Hirasawa[1,3]

1 Laboratory of Pharmacotherapy of Life-Style Related Diseases, Graduate School of Pharmaceutical Sciences, Tohoku University, Aoba-ku, Sendai, Japan, 2 Department of Pharmaceutical Sciences, Tohoku University Hospital, Sendai, Japan, 3 Research Planning and Assessment Office, Tohoku University, Aoba-ku, Sendai, Japan

* ryosuke.segawa.a2@tohoku.ac.jp

## Abstract

Thymic stromal lymphopoietin (TSLP) is an epithelial-derived cytokine that induces type 2 immune responses through dendritic cell activation, and its aberrant regulation is implicated in TSLP-associated inflammatory disorders including atopic dermatitis. We previously demonstrated that hypoxia-inducible factor (HIF) suppresses TSLP expression in human keratinocyte cells; however, the underlying mechanism remained unclear. In this study, we aimed to explore the suppressive mechanism of enarodustat, an HIF-prolyl hydroxylase inhibitor. Enarodustat selectively suppressed TSLP expression induced by the fibroblast-stimulating lipopeptide (FSL-1), a toll like receptor 2/6 agonist in HaCaT, a human keratinocyte cell line. Although both the nuclear factor-κB (NF-κB) and activator protein (AP)-1 contributed to FSL-1-induced TSLP induction, enarodustat preferentially attenuated AP-1 signaling by reducing c-Jun N-terminal kinase (JNK) phosphorylation. This JNK dephosphorylation required both HIF1α and HIF2α and was accompanied by increased expression of dual-specificity phosphatases (DUSPs), which target JNK for dephosphorylation. Collectively, our findings identify a previously uncharacterized HIF–DUSP–JNK axis that negatively regulates TSLP expression. This study provides mechanistic insight into how HIF activation shapes epithelial cytokine responses, offering a basis for understanding the pathogenesis of TSLP-associated diseases such as atopic dermatitis.

## 1. Introduction

Thymic stromal lymphopoietin (TSLP) is an immunoregulatory cytokine derived from epithelial cells. It was initially identified as a lymphocyte growth factor in a thymus-derived fibroblast cell line [1]. Soumelis et al. (2002) reported that TSLP is highly expressed in the inflamed skin of patients with atopic dermatitis [2]. Moreover, TSLP

**Data availability statement:** All relevant data are within the manuscript and its Supporting Information files.

**Funding:** This study was supported by JSPS KAKENHI Grant Number JP21K15252. URL is below. https://www.jsps.go.jp/english/e-grants/ Funders did not play any role in the study design, data collection and analysis, decision to publish, or preparation of the manuscript.

**Competing interests:** The authors have declared that no competing interests exist.

is considered as a key factor contributing to the onset and aggravation of allergic diseases. It also activates the dendritic cells [2], T cells [3], and Group2 innate lymphoid cells (ILC2) [4] and strongly induces type 2 immune responses. TSLP receptor-deficient mice exhibit attenuated allergic immune responses [5], and overproduction of TSLP in the skin causes atopic dermatitis-like symptoms [6]. These findings highlight TSLP as a central mediator in the pathogenesis of TSLP-associated allergic diseases, and controlling TSLP production is considered important for understanding their onset and progression.

TSLP is produced by epithelial cells in response to inflammatory cytokines [7], toll-like receptor ligands [8], and proteases [9]. Nuclear factor (NF)-κB and activator protein (AP)-1, which are activated downstream of the stimuli, play important roles in the transcriptional activation of TSLP [7]. Various nuclear receptors, such as glucocorticoid receptor and retinoid X receptor, negatively regulate TSLP expression by modulating these pathways [10,11]. We recently identified hypoxia-inducible factor (HIF) as an additional negative regulator of TSLP expression in keratinocytes [12]. HIF is a transcription factor that is hydroxylated under normoxic condition by HIF-prolyl hydroxylase (HIF-PH) and degraded by the ubiquitin–proteasome system. In contrast, HIF degradation is suppressed under hypoxic conditions, where HIF exhibits transcriptional activity. Two isoforms of HIF, HIF1α and HIF2α, are expressed in the skin and contributes to the physiological roles of epithelial cells, such as re-epithelialization [13], wound healing [14,15], and production of the barrier protein, filaggrin [16].

Various HIF-PH inhibitors enhancing the transcriptional activity of HIF have been developed and clinically applied as therapeutic agents for renal anemia [17,18]. Beyond hematologic regulation, HIF influences epithelial barrier function [19] and immune responses, including macrophage/T cell activation and cytokine production [20–22]. These observations suggest that HIF signaling may shape epithelial immune responses; however, its mechanistic relationship with TSLP regulation remains incompletely understood.

We previously reported that HIF-PH inhibition reduces TSLP expression in keratinocytes, but the underlying mechanism had not been clarified. In the present study, we investigated how the HIF-PH inhibitor enarodustat regulates TSLP expression and identified a mechanism involving c-Jun N-terminal kinase (JNK) dephosphorylation. These observations prompted us to investigate how HIF-PH inhibition affects TSLP regulation at a mechanistic level, with the aim of elucidating fundamental principles rather than evaluating therapeutic potential.

## 2. Materials and methods

### 2.1. Materials

Enarodustat was provided by Japan Tobacco, Inc. (Tokyo, Japan). Cobalt(II) Chloride ($CoCl_2$) was purchased from Wako Pure Chemical Industries (Osaka, Japan). Dulbecco's modified Eagle's medium (DMEM) was purchased from Nissui Pharmaceutical Co. (Tokyo, Japan). The KGM Gold Bullet Kit was purchased from Lonza (Walkersville, MD, USA). Fetal bovine serum (FBS) was obtained from Biowest

(Miami, FL, USA), and penicillin G, potassium, and streptomycin sulfate were obtained from Meiji Seika Co. (Tokyo, Japan). Human recombinant tumor necrosis factor-alpha (TNF-α) and interleukin (IL)-4 were purchased from Peprotech (Rocky Hill, NJ, USA). Fibroblast-stimulating lipopeptide-1 (FSL-1) was purchased from AdipoGen (San Diego, CA, USA). SLIGKV-NH2, a protease-activated receptor 2 agonist, was purchased from Abcam (Cambridge, UK). Additionally, 2-[(aminocarbonyl)amino]-5-(4-fluorophenyl)-3-thiophenecarboxamide (TPCA-1) was purchased from Selleck Biotechnology (Tokyo, Japan). T-5224 was purchased from Cayman Chemical (Ann Arbor, MI, USA). SP600125 was purchased from Enzo Life Sciences (Farmingdale, NY, USA). Endoribonuclease-prepared siRNA (esiRNA) targeting enhanced green fluorescence protein (EGFP), HIF1A, and EPAS1 (HIF2A) were purchased from Sigma-Aldrich (St. Louis, MO, USA). Lipofectamine RNAiMAX Reagent and Opti-MEM were purchased from Thermo Fisher Scientific (Waltham, MA, USA). Primary antibodies against IκB, NF-κB p65, histone H3, phospho-JNK, phospho-p38, phospho-extracellular signal-regulated kinase (ERK), HIF1α, and glyceraldehyde 3-phosphate dehydrogenase (GAPDH) were purchased from Cell Signaling Technology (Massachusetts, USA). Primary antibody against α-tubulin was purchased from Santa Cruz Biotechnology (Texas, USA). Horseradish peroxidase-conjugated anti-mouse and anti-rabbit IgG antibodies were purchased from Cell Signaling Technology. RNAiso Plus and PrimeScript RT Master Mix were purchased from Takara Bio (Shiga, Japan). KAPA SYBR FAST Universal quantitative polymerase chain reaction (qPCR) Kit was purchased from Nippon Genetics (Tokyo, Japan). LysoPure Nuclear and Cytoplasmic Extractor Kit was purchased from Wako Pure Chemical Industries (Osaka, Japan).

## 2.2. Cell culture

HaCaT cells were kindly provided by Dr. Moriya (Ohu University, Fukushima, Japan). HaCaT cells were maintained in DMEM supplemented with 10% FBS, 18 μg/mL of penicillin G potassium, and 50 μg/mL streptomycin sulfate. HaCaT cells were seeded at a density of $1.5 \times 10^5$ cells/mL in complete KGM medium, containing GA-1000, epinephrine, transferrin, insulin, bovine pituitary extract, hydrocortisone, and hEGF, and incubated for two days. Then, the cells were treated with enarodustat or $CoCl_2$ for 24 h and stimulated with TNF-α (30 ng/mL), PAR2-AP (100 μM), IL-4 (10 ng/mL) and FSL-1 (30 ng/mL) for 30 min for western blotting or 2 h for RT-qPCR analysis. TPCA-1, T-5224, and SP600125 were added simultaneously with FSL-1 stimulation.

## 2.3. RNA extraction and RT-qPCR

RNA extraction, and RT-qPCR were performed as previously described [23]. HaCaT cells were lysed using RNAiso Plus. cDNA sequences (10–30 ng) were amplified using the following primers: human *HPRT1*, 5′-TTGCTTTCCTTGGTCAG GCA-3′ (forward) and 5′-ATCCAACACTTCGTGGGGTC-3′ (reverse); human *GAPDH*, 5′-GAGTCAACGGATTTG GTCGT-3′ (forward) and 5′-CATGGGTGGAATCATATTGGA-3′ (reverse); human *RPL13A*, 5′-GTACGCTGTGAAGG CATCAAC-3′ (forward) and 5′-ACCACCATCCGCTTTTTCTTG-3′ (reverse); human *TSLP*, 5′-GATTACATATATGAGTGG GAC-3′ (forward) and 5′-TTCATTGCCTGAGTAGCAT-3′ (reverse); human *HIF1A*, 5′-TTCACCTGAGCCTAATAGTCC-3′ (forward) and 5′-CAAGTCTAAATCTGTGTCCTG-3′ (reverse); human *HIF2A*, 5′-GTGACATGATCTTTCTGTCGGA-3′ (forward) and 5′-CGCAAGGATGAGTGAAGTCAAA-3′ (reverse); human dual-specificity phosphatase (*DUSP)-1*, 5′-GGCCATTGACTTCATAGACTCC-3′ (forward) and 5′-AACTCAAAGGCCTCGTCCAG-3′ (reverse); human *DUSP2*, 5′-ATAGGCTTCATTGACTGGGTGA-3′ (forward) and 5′-TGCATGAGGTATGCCAGACAG-3′ (reverse); human *DUSP3*, 5′-CTGCCGACTTCATTGACCAG-3′ (forward) and 5′-TAGGCGATAACTAGCGTTGGG-3′ (reverse); human *DUSP4*, 5′-CGGCTCTGTTGAATGTCTCCT-3′ (forward) and 5′-TTCACGGCATCGATGTACTCT-3′ (reverse); human *DUSP5*, 5′-CCTGCTGAATGTCTCCCGAC-3′ (forward) and 5′-GCCTCCCTTTTCCCTGACAC-3′ (reverse); human *DUSP6*, 5′-CTGGAAGGTGGCTTCAGTAAGT-3′ (forward) and 5′-ATTGGGGTCTCGGTCAAGGT-3′ (reverse); human *DUSP7*, 5′-GCCATCAGCTTCATTGACGA-3′ (forward) and 5′-GTAGGCGTCGTTGAGTGACA-3′ (reverse); human *DUSP8*, 5′-TCATCTGCGAGAGCCGCTTCAT-3′ (forward) and 5′-AGCCAGACAGTGGACGATGACT-3′ (reverse); human *DUSP9*,

5′-ATTGAGTTCATTGATGAGGCCT-3′ (forward) and 5′-ACAGTGACGGTGACAGAACG-3′ (reverse); human *DUSP10*, 5′-CAATGAACCAAGCCGAGTGATG-3′ (forward) and 5′-CACAGAGGTTTTCATGGTTCTGC-3′ (reverse); human *DUSP12*, 5′-ATCTATGGCGCCTCTTCGTG-3′ (forward) and 5′-CACTTCGACTGACTCCTGCAT-3′ (reverse); human *DUSP13*, 5′-AGTGTGAGCTACCTGGGGG-3′ (forward) and 5′-CACACAGTGCACCAGGACC-3′ (reverse); human *DUSP14*, 5′-GCACACTGGACTCTTGAGGAA-3′ (forward) and 5′-GAGCAATGCCTCCTATGTCTCC-3′ (reverse); human *DUSP16*, 5′-CCCAAGATGTTGCCTCTCTCT-3′ (forward) and 5′-GGCCAGGGAAACAACGAGA-3′ (reverse); human *DUSP18*, 5′-ATCCACAGCGTGGAGATGAAGC-3′ (forward) and 5′-GCGTGGTACTTCATGAGGTAGG-3′ (reverse); human *DUSP19*, 5′-GGTTGCTCCTAGGGTCACAA-3′ (forward) and 5′-AGGATGTTGGTTTCAGGCAGA-3′ (reverse); human *DUSP22*, 5′-CCATGGGGAATGGGATGAAC-3′ (forward) and 5′-GCACTATCGTGGACAGACAGA-3′ (reverse); human *DUSP23*, 5′-AGATCGTGGACGAGGCCAA-3′ (forward) and 5′-CGCTCCTTCACCAGGTAACA-3′ (reverse). The relative quantities of target mRNAs were determined using the comparative CT (ΔΔCT) method. Primer specificity was confirmed via melting curve analysis. Human *HPRT1*, *GAPDH* and *RPL13A* were used as normalization controls. In Fig 5 and Fig 8, fold change of mRNA expression was calculated and shown as a result. Fold change of mRNA expression was acquired to divide the relative quantities determined from enarodustat treated groups by that determined from control groups. Primers for human *TSLP* were constructed based on the sequence of *Homo sapiens* TSLP transcript variant 1 (long-form TSLP). The CFX Connect Real-Time System (Bio-Rad, California, USA) was used for RT-qPCR.

### 2.4. Western blotting

Western blotting was performed as previously described [23]. Briefly, the protein samples were subjected to 10% (w/v) sodium dodecyl sulfate-polyacrylamide gel electrophoresis (SDS-PAGE) and transferred to the nitrocellulose membranes (GVS Japan, Tokyo, Japan). After blocking with Block Ace (Dainippon Pharmaceutical Co, Tokyo, Japan), the blots were probed with primary antibodies, followed by incubation with HRP-conjugated anti-mouse and anti-rabbit IgG antibodies. Finally, the blots were developed using the ECL chemiluminescence detection system (Cytiva, Marlborough, MA, USA) and chemiluminescence was detected using the ChemiDoc Imaging System (Bio-Rad). Band intensity was quantified using ImageJ software version 1.47 (National Institutes of Health, USA). GAPDH and α-tubulin were used as loading controls.

### 2.5. Fractionation of nuclear proteins

HaCaT cells were seeded in a 60-mm dish and treated with FSL-1 and enarodustat as described above. After treatment, the cells were harvested and fractionated using the LysoPure Nuclear and Cytoplasmic Extractor Kit (Wako Pure Chemical Industries), according to the manufacturer's instructions.

### 2.6 Knockdown of HIF1α and HIF2α expression

Transfection complex was formed using the Lipofectamine RNAiMAX Reagent and Opti-MEM. The esiRNA targeting EGFP was used for the control group. Cells were transfected with the esiRNA targeting EGFP was transfected at 30 pmol/well and esiRNA targeting HIF1α and HIF2α at 15 pmol/well each. Then, the cells were seeded at a density of $1.5 \times 10^5$ cells/mL in complete KGM medium using a 24-well plate, and the transfection complex was simultaneously added. Enarodustat treatment and FSL-1 stimulation were performed as previously described.

### 2.7. Statistical analysis

Excel statistics (version 7.0; ESUMI, Tokyo, Japan) was used for all statistical analyses. Data are expressed as the mean ± standard error of the mean (S.E.M.). A two-tailed paired Student's t-test was used to compare the data between the two groups. Dunnett's test was used for multiple comparisons following analysis of variance (ANOVA).

# 3. Results

## 3.1. Enarodustat inhibits FSL-1-induced TSLP expression

Here, we evaluated the effect of enarodustat on the induction of TSLP mRNA expression using several receptor agonists. Enarodustat did not inhibit TSLP mRNA expression induced by TNF-α (Fig 1A) or protease activated receptor 2-agonist (PAR2-agonist) + IL-4 (Fig 1B). However, enarodustat significantly suppressed TSLP mRNA expression induced by FSL-1, a toll like receptor 2/6 agonist (Fig 1C). Moreover, enarodustat suppressed the FSL-1-induced TSLP expression in a concentration dependent manner (Fig 1D). These results suggest that enarodustat does not broadly suppress TSLP but rather acts on a specific upstream pathway selectively engaged by FSL-1 stimulation. Therefore, we focused on the mechanism of suppression of FSL-1-induced TSLP expression by enarodustat in subsequent experiments.

## 3.2. NF-κB activation is not affected by enarodustat treatment

FSL-1 induces NF-κB and AP-1 activation [24] and these transcription factors contribute to TSLP expression [7,25]. Therefore, we evaluated the effects of enarodustat on FSL-1-induced NF-κB activation in this study. I kappa B Kinase 2 (IKK-2) activation, IκBα degradation and NF-κB p65 nuclear translocation are necessary for NF-κB activation [26]. Here, FSL-1-induced inhibitor of κB alpha (IκBα) degradation (Fig 2A) and NF-κB p65 nuclear translocation (Fig 2B), and these effects were not prevented by enarodustat. We also investigated the synergistic effects of TPCA-1, an IKK-2 inhibitor, and enarodustat. TPCA-1 inhibited FSL-1-induced TSLP mRNA expression in a concentration-dependent manner (Fig 2C). Subsequent enarodustat treatment further and significantly suppressed the FSL-1-induced TSLP mRNA expression (Fig 2D). Thus, enarodustat-mediated inhibition of TSLP expression appears to occur independently of the canonical NF-κB pathway, indicating the involvement of an alternative signaling cascade.

## 3.3. Enarodustat regulates AP-1 activation by suppressing JNK phosphorylation

Next, the effect of enarodustat on AP-1 activation was analyzed. T-5224, a c-Fos/AP-1 inhibitor, partially suppressed FSL-1-induced TSLP mRNA expression in a concentration-dependent manner (Fig 3A). However, the expression that was not inhibited by T-5224 (10 μM) remained unaffected by enarodustat treatment (Fig 3B). We also examined the effects of enarodustat on the phosphorylation of mitogen-activated protein kinases (MAPK) involved in AP-1 activation. FSL-1 induced the phosphorylation of ERK, p38, and JNK. Interestingly, only JNK phosphorylation was inhibited by enarodustat (Fig 3C and 3D). These findings indicate that JNK, rather than ERK or p38, represents a key regulatory node through which enarodustat attenuates AP-1–dependent TSLP expression.

## 3.4. Cobalt chloride suppresses FSL-1-induced TSLP expression and JNK phosphorylation

To confirm the effects of other HIF induction way on FSL-1 induced signaling, we evaluated the effects of cobalt chloride ($CoCl_2$), a classical hypoxia-mimicking agent, on FSL-1-induced TSLP expression and JNK phosphorylation. Corresponding the results of enarodustat, $CoCl_2$ treatment significantly suppressed FSL-1-induced TSLP expression (Fig 4A). $CoCl_2$ treatment strongly induced HIF1α expression and suppressed JNK phosphorylation (Fig 4B and 4C). Consistent effects of enarodustat and $CoCl_2$ suggest that HIF activation is functionally linked to JNK inactivation, positioning HIF upstream of JNK modulation in this context.

## 3.5. JNK inhibitor suppresses FSL-1-induced TSLP expression

To confirm the role of JNK in TSLP induction, we examined the effect of SP600125, a JNK inhibitor, on FSL-1-induced TSLP expression. SP600125 inhibited the FSL-1-induced phosphorylation of JNK (Fig 5A) and significantly suppressed the FSL-1-induced TSLP mRNA expression (Fig 5B). Further, we investigated the contribution of JNK–AP-1 signal

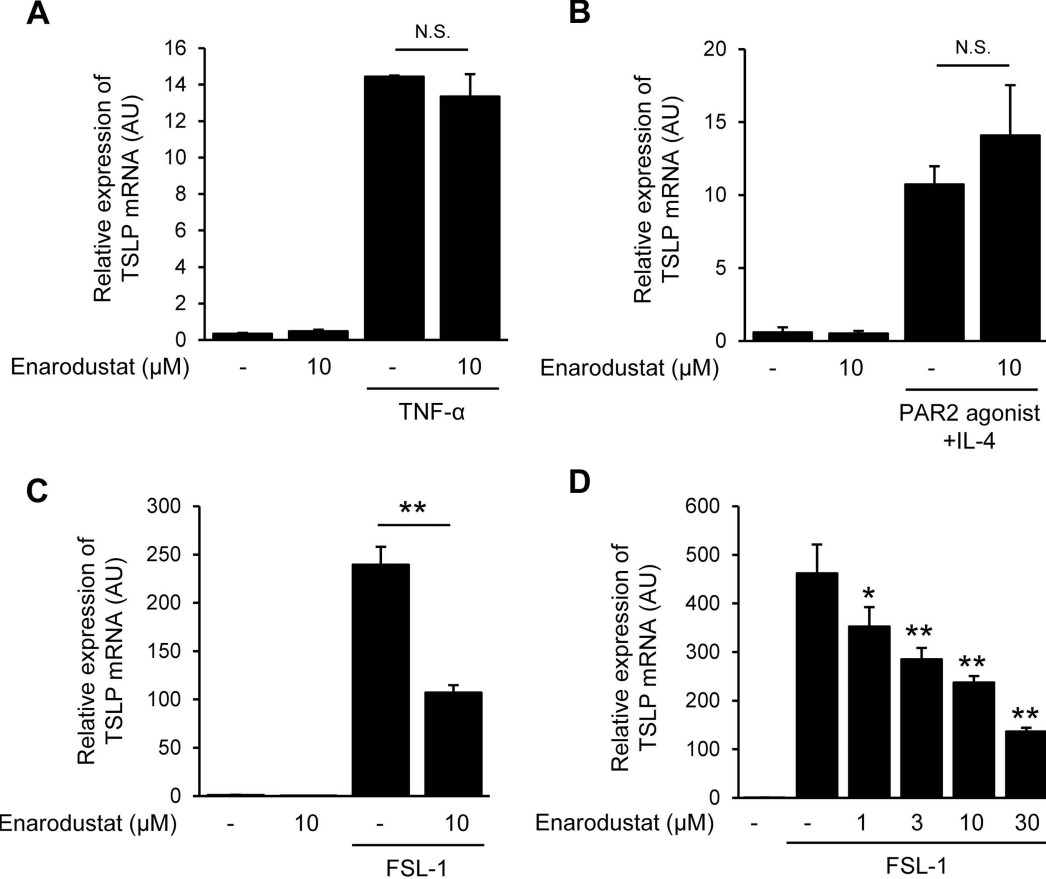

**Fig 1. Enarodustat inhibits FSL-1-induced TSLP expression.** (A-C) TSLP mRNA expression levels in HaCaT cells 2 h after TNF-α (A), PAR2-agonist+IL-4 (B), and FSL-1 (C) stimulation. Cells were treated with enarodustat 24 h before stimulation. mRNA expression was determined by RT-qPCR. (D) Concentration-dependent effect of enarodustat on FSL-1-induced TSLP mRNA expression. *$p < 0.05$, **$p < 0.01$ vs. FSL-1 alone group. Data are represented as the mean ± S.E.M. ($n = 3$). N.S., not significant.

activation on TSLP mRNA expression induced by TNF-α and PAR2-agonist+IL-4. T-5224 did not inhibit TSLP mRNA expression induced by TNF-α (Fig 5C), but significantly inhibited that induced by the PAR2-agonist+IL-4 (Fig 5D). In contrast, SP600125 significantly enhanced the TSLP mRNA expression induced by the PAR2-agonist+IL-4 (Fig 5E). These data indicate that the contribution of JNK–AP-1 signaling to TSLP expression is stimulus-dependent, and FSL-1-driven TSLP expression is uniquely JNK-sensitive compared to other stimuli.

### 3.6. Enarodustat induces JNK-related DUSP family expression

Dual specificity phosphatase (DUSP) family is a representative phosphatase family of MAPKs. To date, 18 types of DUSPs have been reported to reduce JNK phosphorylation [27]. Here, we evaluated the effect of enarodustat on DUSP expression. Among the 18 types of DUSPs, the expression levels of 7 DUSPs were significantly upregulated and only DUSP12 expression was significantly suppressed by enarodustat treatment (Fig 6). Notably, the expression levels of DUSP1, 3, 4, 6, and 9 were increased by more than 1.5-fold by enarodustat. This selective induction of JNK-targeting DUSPs supports a mechanism in which enarodustat drives JNK inactivation through phosphatase-mediated dephosphorylation rather than inhibition of upstream kinase activation.

**A**

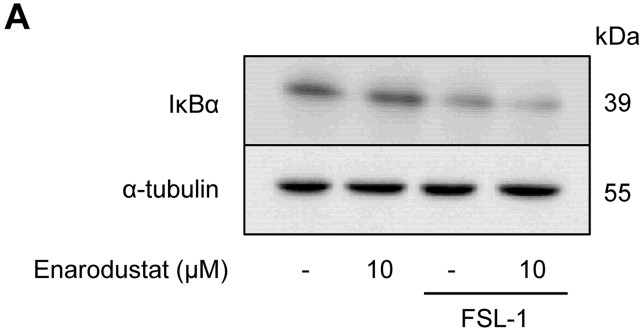

IκBα — 39 kDa

α-tubulin — 55

| Enarodustat (µM) | - | 10 | - | 10 |

FSL-1

**B**

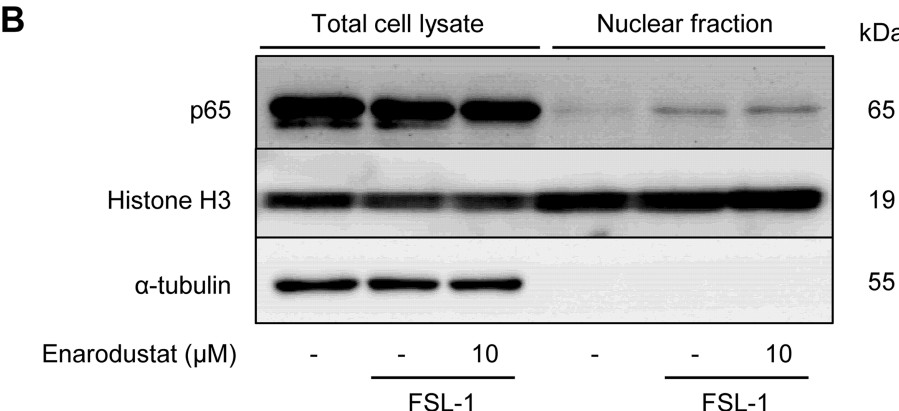

Total cell lysate  Nuclear fraction  kDa

p65 — 65

Histone H3 — 19

α-tubulin — 55

| Enarodustat (µM) | - | - | 10 | - | - | 10 |

FSL-1      FSL-1

**C**

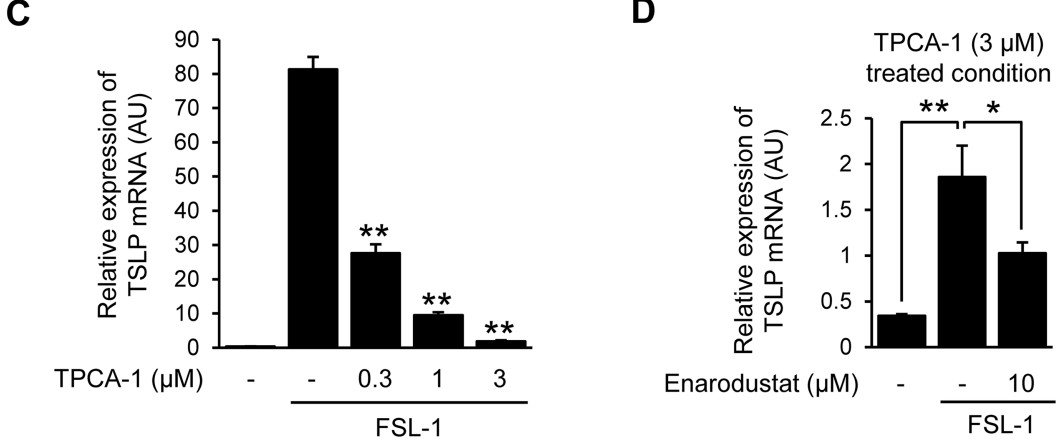

Relative expression of TSLP mRNA (AU)

| TPCA-1 (µM) | - | - | 0.3 | 1 | 3 |

FSL-1

**D**

TPCA-1 (3 µM) treated condition

Relative expression of TSLP mRNA (AU)

| Enarodustat (µM) | - | - | 10 |

FSL-1

**Fig 2. NF-κB activation is not affected by enarodustat treatment.** (A) Expression levels of IκBα after enarodustat and FSL-1 treatment. Samples were collected 30 min after FSL-1 stimulation. IκBα and α-tubulin proteins levels were determined by western blotting. α-tubulin was used as a loading control. (B) Expression NF-κB p65 in nuclear fraction. Histone H3 was used as a loading control for the nuclear fraction samples. α-tubulin was used as a loading control for the total cell lysate. (C) Concentration-dependent effects of TPCA-1 on TSLP mRNA expression in FSL-1-stimulated HaCaT cells. (D) Effects of enarodustat on TSLP mRNA expression after TPCA-1 treatment. *$p < 0.05$, **$p < 0.01$ vs. FSL-1 alone group. Data are represented as the mean ± S.E.M. ($n = 3$). N.S., not significant.

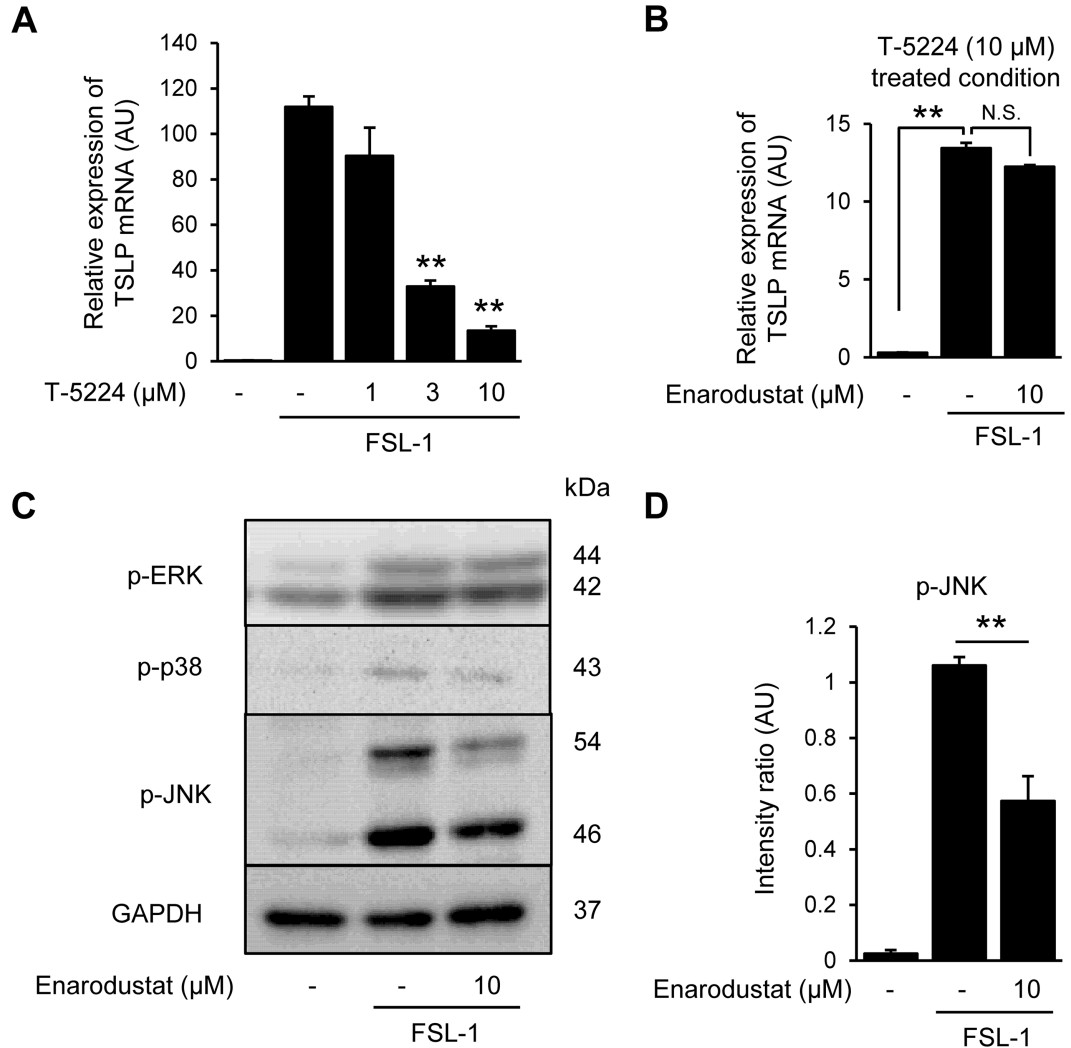

**Fig 3. Enarodustat regulates AP-1 activation by suppressing JNK phosphorylation.** (A) Concentration-dependent effects of T-5224 on TSLP mRNA expression in FSL-1-stimulated HaCaT cells. (B) Effects of enarodustat on TSLP mRNA expression after T-5224 treated. (C) Expression levels of phospho-ERK, p38 and JNK after enarodustat and FSL-1 treatment. Samples were collected 30 min after FSL-1 stimulation. Phospho-MAPKs and GAPDH proteins were determined by western blotting. GAPDH was used as a loading control. (D) Intensity ratio of phospho-JNK/GAPDH. Band intensity was quantified using the ImageJ software. *$p < 0.05$, **$p < 0.01$ vs. FSL-1 alone group. Data are represented as the mean ± S.E.M. ($n = 3$). N.S.; not significant.

### 3.7. Knockdown of HIF attenuates the suppression of TSLP expression by enarodustat

To determine the contribution of HIF to the suppression of TSLP expression by enarodustat, we conducted siRNA knockdown experiments of HIF1α and HIF2α. Combination of HIF1α and HIF2α siRNAs significantly suppressed both HIF1α and HIF2α mRNA expression levels compared with those in the control siRNA-treated group (Fig 7A and 7B). Enarodustat increased the levels of HIF1α; however, HIF1α and HIF2α siRNA transfection inhibited this effect (Fig 7C and 7D). Notably, HIF1α and HIF2α knockdown alone reduced FSL-1-induced TSLP expression, and under this condition, additional enarodustat treatment did not further decrease TSLP levels (Fig 7E). These results indicate that when HIF activity is diminished, the enarodustat-mediated reduction of TSLP is no longer

**A**

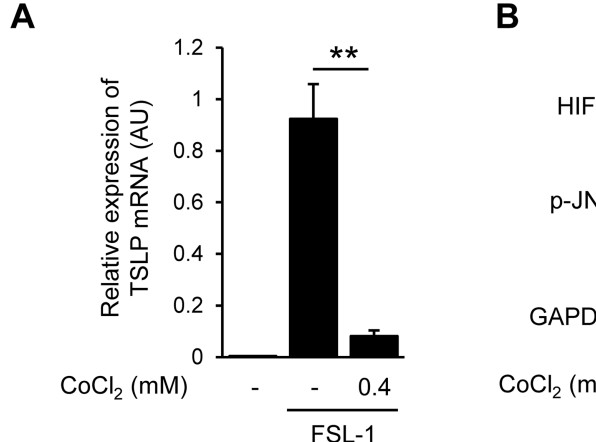

**B**

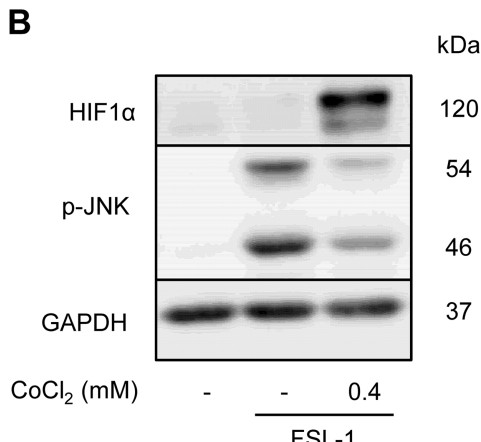

**C**

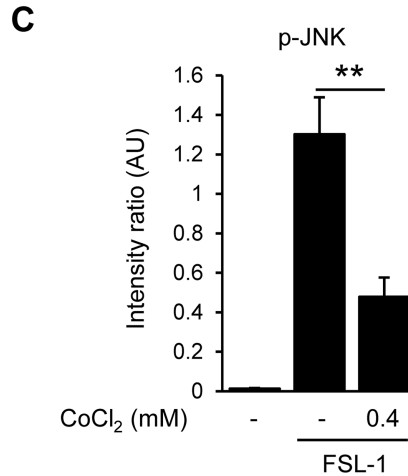

**Fig 4. Cobalt chloride suppresses FSL-1-induced TSLP expression and JNK phosphorylation.** (A) Effects of $CoCl_2$ on FSL-1 induced TSLP mRNA expression. $CoCl_2$ (0.4 mM) was treated 24 h before FSL-1 stimulation in HaCaT cells. (B) Expression levels of HIF1α, phosphor-JNK and GAPDH. Samples were collected 30 min after FSL-1 stimulation. GAPDH was used as a loading control. (C) Intensity ratio of phospho-JNK/GAPDH. Band intensity was quantified using the ImageJ software. *$p < 0.05$, **$p < 0.01$ vs. FSL-1 alone group. Data are represented as the mean ± S.E.M. ($n = 3$).

evident, suggesting that HIF contributes to, but is not the sole determinant of, the suppressive effect on TSLP expression.

### 3.8. Knockdown of HIF expression attenuates JNK dephosphorylation by enarodustat

To confirm the effect of HIF1α and HIF2α knockdown on JNK signaling, we investigated the levels of several DUSPs and JNK phosphorylation after HIF1α and HIF2α knockdown. Enarodustat treatment-induced increase in the mRNA expression levels of DUSP1, 3, 4, 6, 9 in the control siRNA-treated group were significantly inhibited by HIF1α and HIF2α siRNA treatment (Fig 8A). Moreover, the inhibition of FSL-1-induced JNK phosphorylation by enarodustat was blocked by HIF1α and HIF2α siRNA treatments (Fig 8B and 8C). Together, these results indicate that HIF acts as a molecular regulator linking enarodustat treatment to DUSP induction and subsequent JNK dephosphorylation, thereby forming a HIF–DUSP–JNK axis controlling TSLP expression.

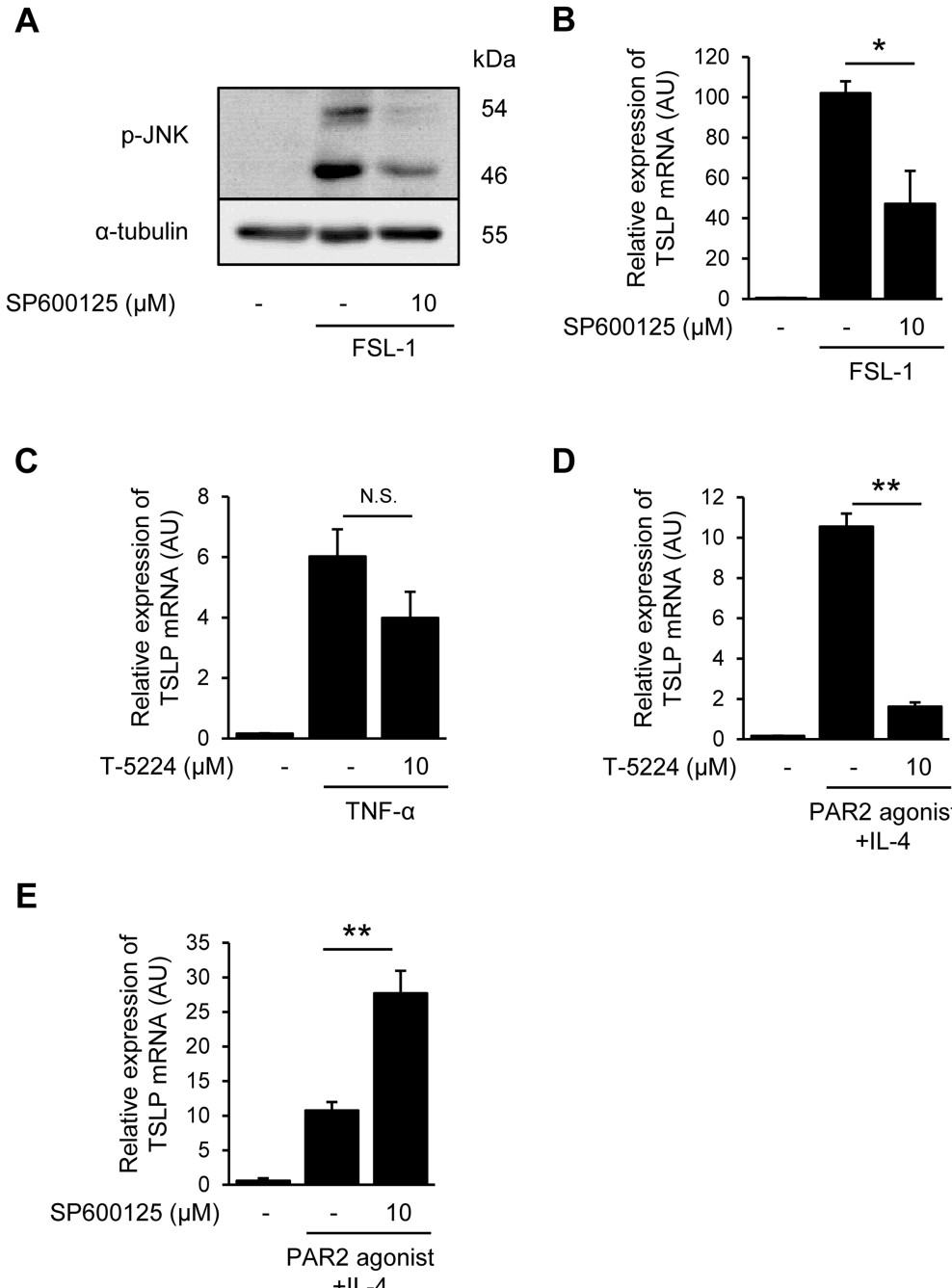

**Fig 5. JNK inhibitor suppresses FSL-1-induced TSLP expression.** (A) Expression levels of phospho-JNK after SP600125 and FSL-1 treatment. Samples were collected 30 min after FSL-1 stimulation. Phospho-JNK and α-tubulin proteins were determined by western blotting. α-tubulin was used as a loading control. (B) Effects of SP600125 on FSL-1 induced TSLP mRNA expression. (C and D) Effects of T-5224 on TSLP mRNA expression levels induced by TNF-α (C) or PAR2-agonist+IL-4 (D). (E) The effects of SP600125 on TSLP mRNA expression induced by PAR2-agonist+IL-4. *$p<0.05$, **$p<0.01$ vs. stimulation alone group. Data are represented as the mean±S.E.M. ($n=3$-4). N.S.; not significant.

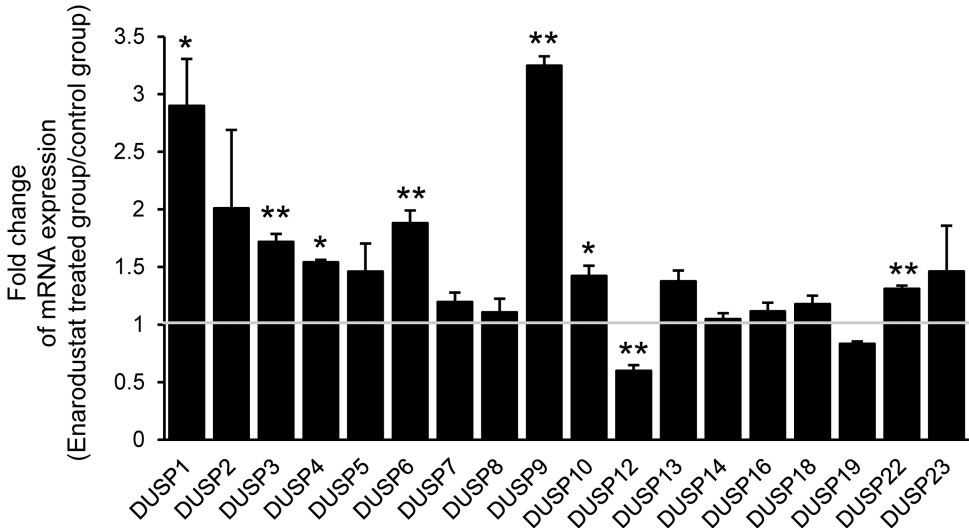

**Fig 6. Enarodustat induces JNK-related DUSP family expression.** DUSP mRNA expression 24 h after enarodustat (10 μM) treatment was determined by RT-qPCR. mRNA expression in the enarodustat-treated group was normalized to that in the untreated control group. Normalized expression was indicated as a fold change in mRNA expression. *$p < 0.05$, **$p < 0.01$ vs. untreated control group. Data are represented as the mean ± S.E.M. ($n = 3$).

## 4. Discussion

In this study, we found that enarodustat inhibited FSL-1-induced TSLP expression by reducing JNK phosphorylation through HIF-mediated induction of DUSPs in human epidermal keratinocytes. Classical hypoxia mimicking agent, $CoCl_2$, also showed same effects as enarodustat, supporting the interpretation that HIF activation in keratinocytes contributes to the downregulation of JNK phosphorylation and TSLP expression at a mechanistic level, rather than indicating direct therapeutic relevance.

Enarodustat suppressed AP-1 activation but not NF-κB activation in FSL-1-stimulated HaCaT cells. NF-κB is activated in hypoxic environments [28]. Furthermore, NF-κB is activated by HIF-PH inhibition through a decrease in HIF-PH-dependent IKKβ hydroxylation [29]. Conversely, HIF1α inhibits NF-κB activation in *Escherichia coli*-infected *Drosophila melanogaster* [30]. However, in the FSL-1-stimulated HaCaT cells, the degradation of IκB and nuclear transport of p65 were not affected by enarodustat, indicating that enarodustat did not affect FSL-1-induced activation of NF-κB. In contrast, phosphorylation of JNK, which phosphorylates c-Jun and activates AP-1, was reduced by enarodustat treatment. These findings suggest that enarodustat selectively inhibits the FSL-1-induced activation of AP-1 by reducing JNK phosphorylation, highlighting a specific regulatory mechanism rather than implying a generalized inhibitory effect across inflammatory pathways.

Here, enarodustat inhibited FSL-1-induced TSLP expression in HaCaT cells. Interestingly, it did not inhibit TSLP expression in TNF-α- or PAR2-agonist + IL-4-treated groups, indicating that its effects are stimulus-dependent and should be interpreted as mechanistic selectivity rather than broad pharmacological suppression. TNF-α-induced TSLP expression, mainly induced by NF-κB [31], was not inhibited by T-5224, a c-fos/AP-1 inhibitor in HaCaT cells. Moreover, PAR2-agonist + IL-4-induced TSLP expression was inhibited by T-5224, but not by SP600125, a JNK inhibitor. Therefore, TNF-α- and PAR2-agonist + IL-4-induced TSLP expression is mediated via pathways different from those involved in FSL-1-induced TSLP expression. Our findings contribute to clarifying the mechanistic distinctions among TSLP-inducing stimuli, and suggest that stimulus-specific regulation should be considered when evaluating TSLP-related pathology. Understanding these differences may provide a foundation for future mechanistic studies in atopic dermatitis rather than direct clinical translation.

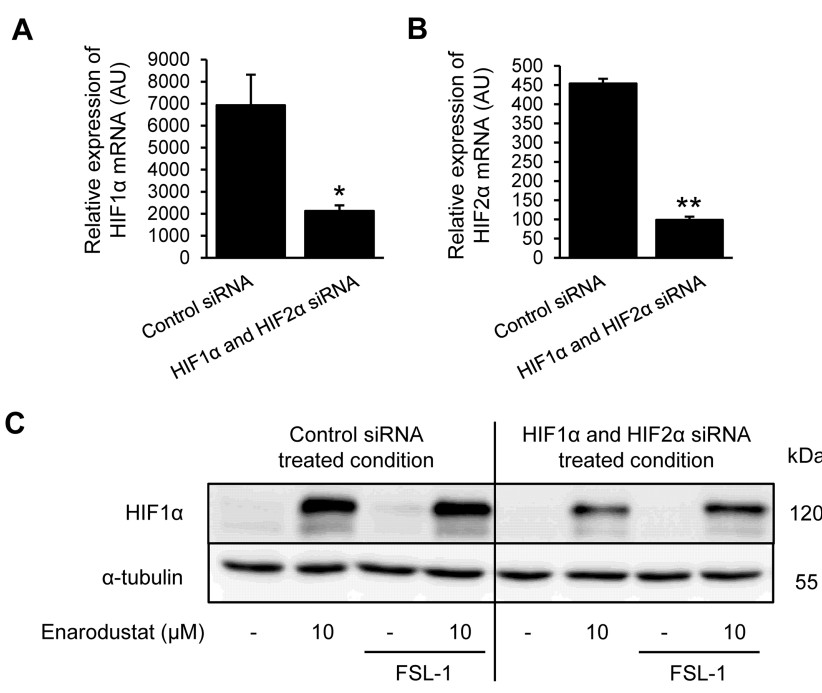

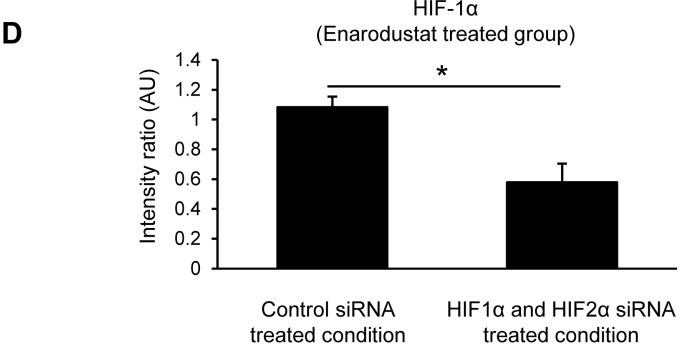

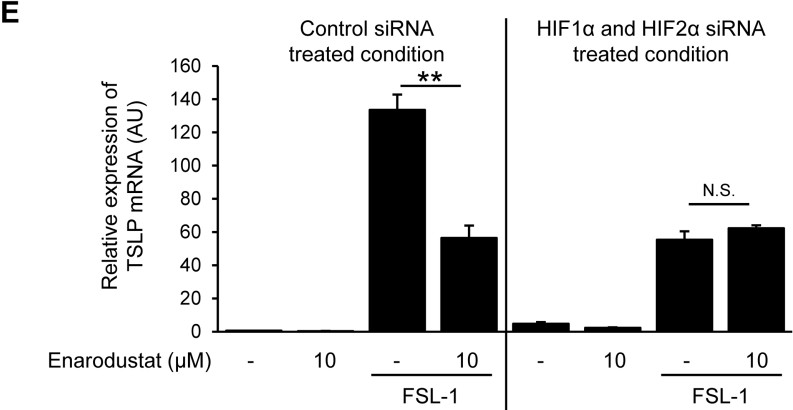

**Fig 7. Knockdown of HIF attenuates the suppression of TSLP expression by enarodustat.** HaCaT cells were transfected with siRNAs at the time of seeding. Cells were treated with enarodustat 24 h before FSL-1 stimulation. (A and B) mRNA expression levels of HIF1α (A) and HIF2α (B) 72 h after siRNA transfection. *$p < 0.05$, **$p < 0.01$ vs. control siRNA-treated group. Data are represented as the mean ± S.E.M. ($n = 3$). (C) Protein expression levels

of HIF1α in enarodustat- and FSL-1-treated groups after control or HIF siRNA treatment. (D) Intensity ratio of HIF1α/α-tubulin in enarodustat treated groups under control siRNA or HIF siRNA treated condition. Band intensity was quantified using the ImageJ software. (E) TSLP mRNA expression levels in control- and HIF siRNA-treated cells. *$p < 0.05$, **$p < 0.01$ vs. FSL-1 alone group. Data are represented as the mean ± S.E.M. ($n = 3$). N.S.; not significant.

FSL-1 is a diacylated lipopeptide that stimulates toll like receptor 2/6 heterodimers and activates downstream signaling molecules. The toll like receptor 2/6 heterodimers activation induced by membrane components of *Staphylococcus aureus* produces TSLP in epidermal keratinocytes [32]. *S. aureus* colonizes lesions in 70% of patients with atopic dermatitis [33]. The amount of toll like receptor 2 ligands in atopic dermatitis lesions correlates with the severity of atopic dermatitis [34]. Moreover, toll like receptor 2 ligand promotes the chronic activation of IL-4-dependent Th2 response in a mouse allergy model [35]. Therefore, the present findings may provide mechanistic insight into how TLR2-mediated signals contribute to TSLP-related inflammation, although any implications for disease modulation remain to be experimentally validated. Furthermore, HIF1α activation in keratinocytes regulates production of anti-bacterial peptides and enhance defensive mechanisms for bacterial infection [36]. Thus, HIF-PH inhibition could conceptually influence multiple aspects of cutaneous immune regulation, but further in vivo analyses will be required to determine pathological and therapeutic relevance. We previously reported that enarodustat inhibits TSLP production and allergen specific IgE production in a mouse allergen sensitization model [12]. However, these findings should be regarded as preliminary supportive observations rather than evidence for clinical utility.

JNK signaling activates HIF in macrophages and epithelial cells [37,38]. JNK activation contributes to the induction of apoptosis by various stimuli, such as UV irradiation [39]. HIF activation exerts anti-apoptotic effects on neurons and pancreatic cancer cells [40,41]. Therefore, HIF is activated downstream of JNK and exerts a negative feedback effect on JNK signaling. Although HIF-PH inhibitors reduce JNK phosphorylation in keratinocytes [42], their action mechanisms remain unclear. DUSP1, 3, 4 directly dephosphorylate JNK [43–45]. Knockdown of DUSP6 enhances the phosphorylation of JNK in T cells [46], and DUSP9 knockdown enhances the phosphorylation of ASK1, which promotes JNK phosphorylation and activity [47]. Induction of DUSP1 expression protects against UV-induced apoptosis by downregulating JNK activation [48]. Therefore, DUSPs regulates JNK activation. In this study, enarodustat induced the expression of DUSPs, particularly DUSP1, 3, 4, 6, and 9. Notably, both DUSP expression and inhibition of TSLP expression by enarodustat were attenuated by HIF1α and HIF2α siRNA treatments. These findings support a mechanistic model in which HIF activation drives DUSP expression to limit JNK signaling, establishing a negative regulatory loop.

Establishing inhibitory environment to JNK activation in epidermal layer of the skin has some advantages for skin disease treatment. JNK regulates cell-cell junction in keratinocytes. JNK inhibition leads to cell adhesion in epidermal keratinocytes via enhancing E-cadherin/β-catenin complex in adherens junction sites [49]. JNK activation prevents appropriate localization of tight junction molecules, such as claudin-4 and ZO-1, and downregulates epithelial barrier function [50]. JNK activation in epidermis of mice leads barrier defects within 1 weeks old and atopic dermatitis like skin inflammation after 3 weeks old [51]. In the context of this study, such information provides mechanistic plausibility but not evidence of clinical benefit, and further basic research will be required to clarify its significance in vivo. Further, HIF regulates filaggrin expression in epithelial tissues [16]. Filaggrin plays an important role in skin barrier function. Thus, HIF activation in epidermal keratinocytes may represent a biological mechanism worth further investigation. Although TSLP protein levels were below the detection limits in both ELISA assays of culture supernatants and western blot analyses under our experimental conditions, TSLP mRNA expression was used as a sensitive and reliable readout to investigate the transcriptional regulatory mechanisms downstream of HIF activation. Accordingly, the present study should be interpreted primarily as a mechanistic analysis of intracellular signaling pathways rather than an evaluation of protein-level regulation or therapeutic efficacy.

**A**

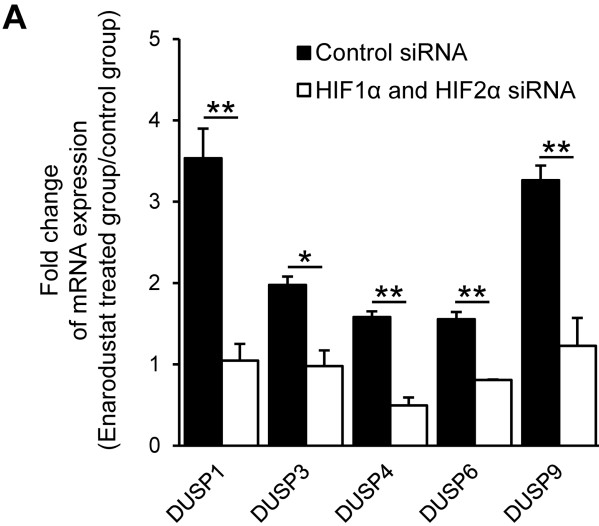

**B**

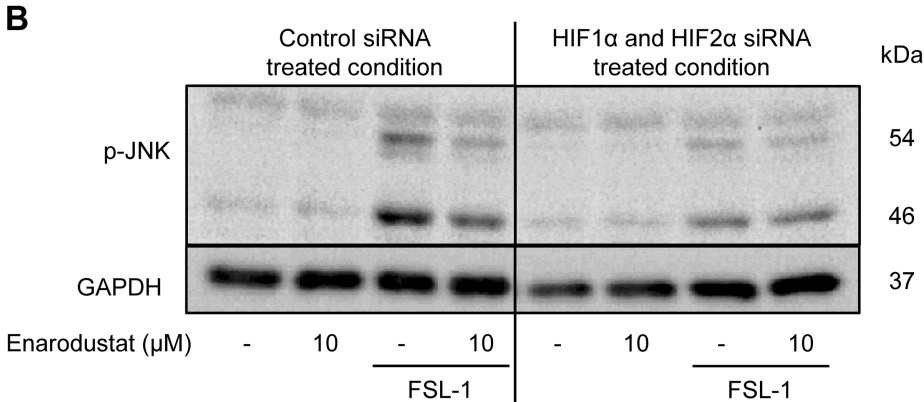

**C**

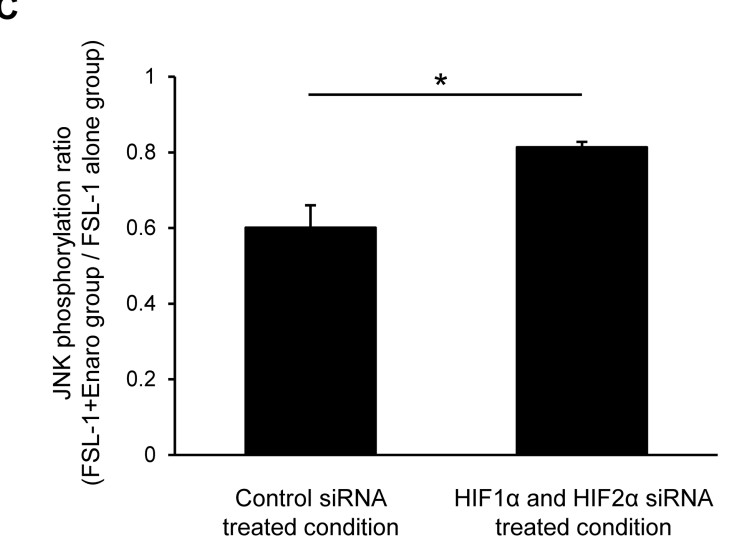

**Fig 8. Knockdown of HIF attenuates JNK dephosphorylation by enarodustat.** (A) Fold change in DUSP1, 3, 4, 6, 9 mRNA expression levels 24 h after enarodustat treatment in the control- (black column) or HIF siRNA- (white column) treated group. mRNA expression in the enarodustat-treated groups was normalized to that in the untreated control group. Normalized expression was indicated as fold change in mRNA expression.. (B) Protein

expression levels of phospho-JNK protein in enarodustat- and FSL-1-treated groups after control or HIF siRNA transfection. (C) Intensity ratio of phospho-JNK/GAPDH in FSL-1 alone groups and FSL-1+enarodustat treated groups was quantified using the ImageJ software. JNK phosphorylation ratio was calculated to devide the intentisy ratio of FSL-1+enarodustat group by that of FSL-1 alone group. $*p < 0.05$, $**p < 0.01$ vs. corresponding control siRNA-treated group. Data are represented as the mean±S.E.M. ($n = 3$).

In summary, this study provides mechanistic evidence that enarodustat suppresses TSLP production induced by TLR2 stimulation in human keratinocytes through HIF-mediated JNK dephosphorylation. These findings reveal a regulatory pathway linking HIF activation, DUSP induction, and JNK signaling suppression, and should be interpreted as foundational mechanistic insight. Considering our findings and previously reported effects of HIF activation and JNK inhibition on barrier function and bacterial infection in the skin, our results suggest a potential basis for future mechanistic exploration of HIF-PH inhibitors in allergic skin inflammation. In addition, comprehensive transcriptomic approaches such as RNA-seq or genome-wide profiling will be considered in future studies to investigate how enarodustat-mediated HIF activation reprograms keratinocyte signaling at a global level. These analyses may help to identify broader HIF-dependent regulatory pathways contributing to TSLP suppression, extending beyond the JNK/AP-1 axis identified in the present work. We believe such an approach will further strengthen the mechanistic foundation of this study and support future translational investigation. Future studies, including in vivo analyses and pathological relevance assessment, will be essential to determine whether these mechanistic findings can inform preventive or therapeutic strategies.

## Supporting information

**S1 raw images. Contains the Original western blot data.**
(PDF)

**S1 Data set. Raw data for all quantitative figures.** This file contains the raw numerical data used for all quantitative analyses presented in Fig 1–8.
(XLSX)

## Acknowledgments
We would like to thank Editage (www.editage.com) for English language editing.

## Author contributions

**Conceptualization:** Ryosuke Segawa.

**Data curation:** Ryosuke Segawa.

**Formal analysis:** Ryosuke Segawa.

**Funding acquisition:** Ryosuke Segawa.

**Investigation:** Ryosuke Segawa, Makiko Yagisawa, Chihiro Miyata.

**Methodology:** Ryosuke Segawa.

**Supervision:** Noriyasu Hirasawa.

**Writing – original draft:** Ryosuke Segawa.

**Writing – review & editing:** Noriyasu Hirasawa.

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
