## [Decision Letter · Decision Letter 0]

1 Dec 2025

Dear Dr. Segawa,

Thank you for submitting your manuscript to PLOS ONE. After careful consideration, we feel that it has merit but does not fully meet PLOS ONE’s publication criteria as it currently stands. Therefore, we invite you to submit a revised version of the manuscript that addresses the points raised during the review process.

We look forward to receiving your revised manuscript.

Kind regards,

Srinivasa Reddy Bonam

Academic Editor

PLOS ONE

Journal Requirements:

4. Please expand the acronym “JSPS” (as indicated in your financial disclosure) so that it states the name of your funders in full.

Reviewers' comments:

Reviewer's Responses to Questions

**Comments to the Author**

1. Is the manuscript technically sound, and do the data support the conclusions?

Reviewer #1: No

Reviewer #2: Yes

2. Has the statistical analysis been performed appropriately and rigorously?

Reviewer #1: Yes

Reviewer #2: Yes

3. Have the authors made all data underlying the findings in their manuscript fully available?

Reviewer #1: Yes

Reviewer #2: Yes

4. Is the manuscript presented in an intelligible fashion and written in standard English?

Reviewer #1: Yes

Reviewer #2: Yes

Reviewer #1: Please find attached my review of the manuscript entitled "Enarodustat suppresses thymic stromal lymphopoietin expression via hypoxia-inducible factor-mediated c-Jun N-terminal kinases dephosphorylation" (manuscript ref no.: PONE-D-25-54477). The authors have already reported in the past that HIF signaling suppresses TSLP expression in keratinocytes, but the mechanism was unclear. In the current study they identify AP-1 (via JNK) as the key suppressed pathway. They also demonstrate that HIF activation increases DUSP expression, which in turn dephosphorylates and inactivates JNK, leading to reduced AP-1 activity and both HIF1α and HIF2α are required for this suppressive effect. While the study suggests use of small molecule enarodustat as a clinically attractive target for Atopic Dermatitis by targeting endogenous HIF pathway, the data are weak and has insufficient proof of concept. The use of a single cell line HaCaT to prove clinical translation is merely speculative. Mechanistic finding requires in vivo validation using AD mouse model or knock outs for DUSP and/or HIF1α/HIF2α. Alternatively in order to investigate how enarodustat and HIF activation reprogram keratinocyte pathways the authors could design studies to investigate transcriptomic and genomic profiles to investigate the pathways in enarodustat-mediated TSLP suppression. This would identify global transcriptional changes in response to enarodustat or Determine HIF-dependent pathways that mediate TSLP suppression.

Reviewer #2: Paper: Enarodustat suppresses thymic stromal lymphopoietin expression via hypoxia inducible factor-mediated c-Jun N-terminal kinases dephosphorylation

Summary and Overall impression:

In this manuscript, Segawa et al. determined the components of pathways involved FSL-1 induction of TSLP within an in vitro system using the human keratinocyte cell line, HaCaT cells. The overall study attempts to systematically determine the involvement of NF-kB and AP-1 pathways in the FSL-1 induction of TSLP through use the of inhibitors. In doing so, they provide a schematic for a pathway in which Enarodustat acts to suppress TSLP through the involvement of stabilization of HIF, and the subsequent dephosphorylation of JNK through a proposed number of DUSPs. While the study has mostly delineated the involvement of the pathways it hypothesized, several concerns arise when reading the text that should be addressed:

Discussion of specific areas for improvement

Major concern

1. Throughout the paper, only RT-qPCR was used for TSLP detection. Given that this paper is arguing for pharmacological use of that HIF-PH inhibitors such as enarodustat in treating atopic dermatitis, have the authors considered using a protein detection method such as ELISA or western blot to ascertain that similar dynamics of TSLP are observed at the protein level? If not, this should be done for Figures 2D and 3B where they attempt to deduce the involvement of enarodustat in the NF-kB and AP-1 pathways.

Minor concern

1. In the results section, authors tend to leave things hanging by solely describing the results without providing suggestions on what they mean. For example, the entire section 3.3 is purely descriptive, without any interpretation on what the data means. This problem is persistent throughout the manuscript. This may have been intended to save the inference for the discussion, but the paragraphs end abruptly, disrupting the understanding of the reader as we do not know the key point that the authors want us to get from the figure.

2. The phrasing in line 262 and 263 needs to be reworked. From Figure 7E, it seems “HIF1a and HIF2a knockdown without Enarodustat” already suppressed TSLP production to a similar level as with the “control siRNA treated condition + Enarodustat”. But the term block gives a false interpretation of the data.

**Do you want your identity to be public for this peer review?** For information about this choice, including consent withdrawal, please see our Privacy Policy

Reviewer #1: No

Reviewer #2: No

---

## [Author Response · Author response to Decision Letter 1]

6 Jan 2026

Response to Reviewers

We sincerely thank the reviewers for their constructive comments and insightful suggestions. We have carefully revised the manuscript to clarify the study’s position as a mechanism-focused investigation rather than a therapeutic proof-of-concept, and to address all points raised. Below, we provide point-by-point responses.

To reviewer 1

Comment 1: While the study suggests use of small molecule enarodustat as a clinically attractive target for Atopic Dermatitis by targeting endogenous HIF pathway, the data are weak and has insufficient proof of concept. The use of a single cell line HaCaT to prove clinical translation is merely speculative. Mechanistic finding requires in vivo validation using AD mouse model or knock outs for DUSP and/or HIF1α/HIF2α.

Response: We appreciate this important perspective. To avoid overstating the findings, we revised the manuscript to emphasize that the present work is positioned as a mechanism-oriented study. Our main objective is to clarify how HIF activation regulates TSLP transcriptional control through DUSPs and JNK dephosphorylation, rather than to assert direct therapeutic efficacy. We acknowledge that validation in additional systems, such as in vivo AD models or genetic loss-of-function approaches (e.g., DUSP or HIF knockout), will be necessary to extend these findings toward translational implications. These points are now clearly stated in the Discussion section.

We have added the paragraph in page 2, lines 29-35: This JNK dephosphorylation required both HIF1α and HIF2α and was accompanied by increased expression of dual-specificity phosphatases (DUSPs), which target JNK for dephosphorylation. Collectively, our findings identify a previously uncharacterized HIF–DUSP–JNK axis that negatively regulates TSLP expression. This study provides mechanistic insight into how HIF activation shapes epithelial cytokine responses, offering a basis for understanding the pathogenesis of TSLP-associated diseases such as atopic dermatitis.

We have added the paragraph in page 3, lines 47-49: These findings highlight TSLP as a central mediator in the pathogenesis of TSLP-associated allergic diseases, and controlling TSLP production is considered important for understanding their onset and progression.

in page 4, lines 63-73: Beyond hematologic regulation, HIF influences epithelial barrier function [20] and immune responses, including macrophage/T cell activation and cytokine production [21–23]. These observations suggest that HIF signaling may shape epithelial immune responses; however, its mechanistic relationship with TSLP regulation remains incompletely understood.

We previously reported that HIF-PH inhibition reduces TSLP expression in keratinocytes, but the underlying mechanism had not been clarified. In the present study, we investigated how the HIF-PH inhibitor enarodustat regulates TSLP expression and identified a mechanism involving c-Jun N-terminal kinase (JNK) dephosphorylation. These observations prompted us to investigate how HIF-PH inhibition affects TSLP regulation at a mechanistic level, with the aim of elucidating fundamental principles rather than evaluating therapeutic potential.

in page 16, lines 295-297: supporting the interpretation that HIF activation in keratinocytes contributes to the downregulation of JNK phosphorylation and TSLP expression at a mechanistic level, rather than indicating direct therapeutic relevance.

in page 17, lines 310-312: indicating that its effects are stimulus-dependent and should be interpreted as mechanistic selectivity rather than broad pharmacological suppression.

in page 18, lines 327-330: Therefore, the present findings may provide mechanistic insight into how TLR2-mediated signals contribute to TSLP-related inflammation, although any implications for disease modulation remain to be experimentally validated.

in page 18, lines 331-333: Thus, HIF-PH inhibition could conceptually influence multiple aspects of cutaneous immune regulation, but further in vivo analyses will be required to determine pathological and therapeutic relevance.

in page 18, lines 335-336: However, these findings should be regarded as preliminary supportive observations rather than evidence for clinical utility.

in page 19, lines 349-351: These findings support a mechanistic model in which HIF activation drives DUSP expression to limit JNK signaling, establishing a negative regulatory loop.

in page 19, lines 358-360: In the context of this study, such information provides mechanistic plausibility but not evidence of clinical benefit, and further basic research will be required to clarify its significance in vivo.

in page 20, lines 370-374: In summary, this study provides mechanistic evidence that enarodustat suppresses TSLP production induced by TLR2 stimulation in human keratinocytes through HIF-mediated JNK dephosphorylation. These findings reveal a regulatory pathway linking HIF activation, DUSP induction, and JNK signaling suppression, and should be interpreted as foundational mechanistic insight.

Comment 2: Alternatively, to investigate how enarodustat and HIF activation reprogram keratinocyte pathways, the authors could design transcriptomic or genomic profiling studies to identify global transcriptional changes and HIF-dependent pathways related to TSLP suppression.

Response: We thank the reviewer for this valuable suggestion. We fully agree that transcriptomic and/or genomic profiling would provide important insights into how enarodustat-mediated HIF activation reprograms keratinocyte signaling and may clarify the global regulatory landscape underlying TSLP suppression. However, the current study was designed as a mechanism-focused investigation, aiming to identify the key signaling pathway responsible for TSLP suppression, rather than to comprehensively evaluate global transcriptional changes. Given this scope, we prioritized pathway-level analyses centered on JNK/AP-1 regulation and DUSP induction.

That said, we recognize the importance of the reviewer’s recommendation. We have now added a statement in the Discussion indicating that transcriptomic approaches (e.g., RNA-seq) will be incorporated into future studies to explore HIF-dependent pathway reprogramming and to validate the broader biological impact of enarodustat. We believe this will strengthen the translational value of our research while maintaining the mechanistic focus of the current work.

We have added the paragraph in page 19-20, lines 360-361: Thus, HIF activation in epidermal keratinocytes may represent a biological mechanism worth further investigation.

in page 20, lines 375-385: our results suggest a potential basis for future mechanistic exploration of HIF-PH inhibitors in allergic skin inflammation. In addition, comprehensive transcriptomic approaches such as RNA-seq or genome-wide profiling will be considered in future studies to investigate how enarodustat-mediated HIF activation reprograms keratinocyte signaling at a global level. These analyses may help to identify broader HIF-dependent regulatory pathways contributing to TSLP suppression, extending beyond the JNK/AP-1 axis identified in the present work. We believe such an approach will further strengthen the mechanistic foundation of this study and support future translational investigation. Future studies, including in vivo analyses and pathological relevance assessment, will be essential to determine whether these mechanistic findings can inform preventive or therapeutic strategies.

To reviewer 2

Major concern 1: Throughout the paper, only RT-qPCR was used for TSLP detection. Given that this paper is arguing for pharmacological use of that HIF-PH inhibitors such as enarodustat in treating atopic dermatitis, have the authors considered using a protein detection method such as ELISA or western blot to ascertain that similar dynamics of TSLP are observed at the protein level? If not, this should be done for Figures 2D and 3B where they attempt to deduce the involvement of enarodustat in the NF-kB and AP-1 pathways.

Response: We attempted to detect secreted TSLP protein using ELISA and assess intracellular protein by western blot; however, in our culture conditions, TSLP remained below detectable limits despite FSL‑1 stimulation. We have added text clarifying this in the Results and Discussion. Therefore, the present study focuses on defining upstream regulatory mechanisms driving transcriptional suppression (HIF–DUSP–JNK axis). We revised statements that could be interpreted as claiming therapeutic efficacy and instead describe the work as mechanistic.

We have added the paragraph in page 20, lines 362-369: Thus, HIF activation in epidermal keratinocytes may represent a biological mechanism worth further investigation. Although TSLP protein levels were below the detection limits in both ELISA assays of culture supernatants and western blot analyses under our experimental conditions, TSLP mRNA expression was used as a sensitive and reliable readout to investigate the transcriptional regulatory mechanisms downstream of HIF activation. Accordingly, the present study should be interpreted primarily as a mechanistic analysis of intracellular signaling pathways rather than an evaluation of protein-level regulation or therapeutic efficacy.

Minor concern 1: In the results section, authors tend to leave things hanging by solely describing the results without providing suggestions on what they mean. For example, the entire section 3.3 is purely descriptive, without any interpretation on what the data means. This problem is persistent throughout the manuscript. This may have been intended to save the inference for the discussion, but the paragraphs end abruptly, disrupting the understanding of the reader as we do not know the key point that the authors want us to get from the figure.

Response: We sincerely thank the reviewer for this constructive suggestion. We agree that several paragraphs in the Results section were overly descriptive and did not sufficiently articulate the key implications of each experiment. To address this concern, we revised the Results section to include brief interpretative statements at the end of relevant paragraphs,

These revisions ensure smoother logical flow and prevent paragraphs from ending abruptly. We believe these changes enhance clarity and improve the structural consistency of the Results section.

We have added the paragraph in page 11, lines 205-207: These results suggest that enarodustat does not broadly suppress TSLP but rather acts on a specific upstream pathway selectively engaged by FSL-1 stimulation.

in page 12, lines 220-222: Thus, enarodustat-mediated inhibition of TSLP expression appears to occur independently of the canonical NF-κB pathway, indicating the involvement of an alternative signaling cascade.

in page 13, lines 231-233: These findings indicate that JNK, rather than ERK or p38, represents a key regulatory node through which enarodustat attenuates AP-1–dependent TSLP expression.

in page 13, lines 241-242: Consistent effects of enarodustat and CoCl₂ suggest that HIF activation is functionally linked to JNK inactivation, positioning HIF upstream of JNK modulation in this context.

in page 14, lines 252-255: These data indicate that the contribution of JNK–AP-1 signaling to TSLP expression is stimulus-dependent, and FSL-1-driven TSLP expression is uniquely JNK-sensitive compared to other stimuli.

in page 14-15, lines 264-266: This selective induction of JNK-targeting DUSPs supports a mechanism in which enarodustat drives JNK inactivation through phosphatase-mediated dephosphorylation rather than inhibition of upstream kinase activation.

in page 15, lines 276-278: These results indicate that when HIF activity is diminished, the enarodustat-mediated reduction of TSLP is no longer evident, suggesting that HIF contributes to, but is not the sole determinant of, the suppressive effect on TSLP expression.

in page 16, lines 286-289: Together, these results indicate that HIF acts as a molecular regulator linking enarodustat treatment to DUSP induction and subsequent JNK dephosphorylation, thereby forming a HIF–DUSP–JNK axis controlling TSLP expression.

Minor concern 2: The phrasing in line 262 and 263 needs to be reworked. From Figure 7E, it seems “HIF1a and HIF2a knockdown without Enarodustat” already suppressed TSLP production to a similar level as with the “control siRNA treated condition + Enarodustat”. But the term block gives a false interpretation of the data.

Response: We thank the reviewer for this insightful comment and agree that the original wording in lines 262–263 could lead to an overinterpretation of the data. As correctly pointed out, Figure 7E shows that knockdown of HIF1α and HIF2α alone already reduced FSL-1–induced TSLP expression to a level comparable to that observed in control siRNA–treated cells with enarodustat, and that additional enarodustat treatment did not further suppress TSLP expression under HIF-deficient conditions.

To address this concern, we have revised the Results section to more accurately describe the experimental observations without implying complete blockade. The revised text now states that enarodustat-mediated suppression of TSLP expression is no longer evident when HIF activity is diminished, indicating that HIF contributes to, but is not the sole determinant of, the suppressive effect on TSLP expression (Fig. 7E).

In addition, we removed the term “block” and adjusted the wording throughout the manuscript to avoid overstating causality, thereby ensuring that the interpretation remains consistent with the data.

We have added the paragraph in page 15, lines 274-278: Notably, HIF1α and HIF2α knockdown alone reduced FSL-1 induced TSLP expression, and under this condition, additional enarodustat treatment did not further decrease TSLP levels (Fig. 7E). These results indicate that when HIF activity is diminished, the enarodustat-mediated reduction of TSLP is no longer evident, suggesting that HIF contributes to, but is not the sole determinant of, the suppressive effect on TSLP expression.

Other revised points are as below.

We have added the sentence in page 2, lines 20-21: and its aberrant regulation is implicated in TSLP-associated inflammatory disorders including atopic dermatitis.

We have revised the sentence in page 2, lines 28-29:

FSL-1-induced TSLP expression, enarodustat preferentially attenuated AP-1 signaling by downregulating the c-Jun N-terminal kinase (JNK) phosphorylation.

➡ TSLP induction, enarodustat preferentially attenuated AP-1 signaling by reducing c-Jun N-terminal kinase (JNK) phosphorylation.

We have added the sentence in page 3, lines 54-56: negatively regulate TSLP expression by modulating these patyways [11,12]. We recently identified hypoxia-inducible factor (HIF) as an additional negative regulator of TSLP expression in keratinocytes [13].

We have added the sentence in page 7, lines 121-122: human RPL13A, 5ʹ-GTACGCTGTGAAGGCATCAAC-3ʹ (forward) and 5ʹ-ACCACCATCCGCTTTTTCTTG-3ʹ (reverse)

We have revised the sentence in page 15, lines 268:

3.7 Knockdown of HIF blocks the suppression of TSLP expression by enarodustat

➡ 3.7 Knockdown of HIF attenuates the suppression of TSLP expression by enarodustat

We have revised the sentence in page 15, lines 280:

3.8 Knockdown of HIF expression blocks JNK dephosphorylation by enarodustat

➡ 3.8 Knockdown of HIF expression attenuates JNK dephosphorylation by enarodustat

We have added the sentence in page 17, lines 307-308: highlighting a specific regulatory mechanism rather than implying a generalized inhibitory effect across inflammatory pathways.

We have added the sentence in page 21, lines 391-392: This study was supported by the Japan Society for the Promotion of Science (JSPS) KAKENHI, Grant Number JP21K15252.

Further we have added the caption of supporting information in page 32-33, line 637-664

---

## [Editor Report · Decision Letter 1]

9 Jan 2026

Enarodustat suppresses thymic stromal lymphopoietin expression via hypoxia-inducible factor-mediated c-Jun N-terminal kinases dephosphorylation

PONE-D-25-54477R1

Dear Prof. Segawa,

Thank you for submitting the revised manuscript.

We’re pleased to inform you that your manuscript has been judged scientifically suitable for publication and will be formally accepted for publication once it meets all outstanding technical requirements.

Kind regards,

Srinivasa Reddy Bonam

Academic Editor

PLOS One

---

## [Editor Report · Acceptance letter]

PONE-D-25-54477R1

PLOS One

Dear Dr. Segawa,

I'm pleased to inform you that your manuscript has been deemed suitable for publication in PLOS One. Congratulations! Your manuscript is now being handed over to our production team.

Kind regards,

on behalf of

Dr. Srinivasa Reddy Bonam

Academic Editor

PLOS One